# Parkinsonisms and Glucocerebrosidase Deficiency: A Comprehensive Review for Molecular and Cellular Mechanism of Glucocerebrosidase Deficiency

**DOI:** 10.3390/brainsci9020030

**Published:** 2019-02-01

**Authors:** Emilia M. Gatto, Gustavo Da Prat, Jose Luis Etcheverry, Guillermo Drelichman, Martin Cesarini

**Affiliations:** 1Department of Neurology, Parkinson’s Disease and Movement Disorders Section, Institute of Neuroscience of Buenos Aires (INEBA), Guardia Vieja 4435, Buenos Aires C1192AAW, Argentina; gustavoda_prat@hotmail.com (G.D.P.); jletcheverry_1@yahoo.com.ar (G.L.E.); martincesarini23@gmail.com (M.C.); 2Hospital de Niños Ricardo Gutiérrez, Gallo 1330, Buenos Aires C1425EFD, Argentina; drgdrelichman@yahoo.com.ar

**Keywords:** glucocerebrosidase, Parkinson’s disease, Gaucher disease

## Abstract

In the last years, lysosomal storage diseases appear as a bridge of knowledge between rare genetic inborn metabolic disorders and neurodegenerative diseases such as Parkinson’s disease (PD) or frontotemporal dementia. Epidemiological studies helped promote research in the field that continues to improve our understanding of the link between mutations in the glucocerebrosidase (*GBA*) gene and PD. We conducted a review of this link, highlighting the association in *GBA* mutation carriers and in Gaucher disease type 1 patients (GD type 1). A comprehensive review of the literature from January 2008 to December 2018 was undertaken. Relevance findings include: (1) There is a bidirectional interaction between GBA and α- synuclein in protein homeostasis regulatory pathways involving the clearance of aggregated proteins. (2) The link between GBA deficiency and PD appears not to be restricted to α–synuclein aggregates but also involves *Parkin* and *PINK1* mutations. (3) Other factors help explain this association, including early and later endosomes and the lysosomal-associated membrane protein 2A (LAMP-2A) involved in the chaperone-mediated autophagy (CMA). (4) The best knowledge allows researchers to explore new therapeutic pathways alongside substrate reduction or enzyme replacement therapies.

## 1. Introduction

In the past decade, advances in the knowledge of the pathophysiological process of Parkinson’s disease (PD) have shed light on the comprehensive mechanisms involved in protein accumulation and aggregation in neurodegenerative diseases [1,2,3].

Parkinson’s is the second most common neurodegenerative disorder. Criteria for diagnosis were redefined in 2015 by Postuma et al., with multiple genes identified as causative or as an increased risk factor [2].

In PD, a complex underlying physiopathology involving molecular processes results in alpha synuclein (α-syn) misfolding, leading abnormal aggregation and the accumulation of insoluble α-syn.

A key player in this matter is the lysosomal-autophagy system (LAS), being an important target for many new therapeutic targets in current clinical trials [1].

New genetic findings help support a concomitant dysfunctional proteostasis involving several systems, including the ubiquitin–proteasome system (UPS), chaperones, and the LAS [1,4,5].

The major candidate gene involved in PD with autosomal dominant inheritance are leucine-rich repeat kinase 2 (*LRRK2*), alpha-synuclein (*SNCA*), vacuolar sorting protein 35 (*VPS35*), and DnaJ homolog subfamily C member 13 (*DNAJC13*). In the recessive pattern the major genes players are *Parkin*, *PINK1*, and *DJ1*, as well as *GBA* [6,7,8].

Although the understanding of these gene functions is incomplete, several of them are involved in different protein and organelle clearance pathways.

In this sense, the homeostasis of α-synuclein depends on the ubiquitin–proteasome system (UBQ-PS) and the LAS that comprise the chaperone-mediated autophagy and macroautophagy [9]. The *SNCA* mutations and multiplications promote the accumulation of α-synuclein oligomers inhibiting the UBQ-PS and macroautophagy.

The *LRRK2* gene encodes a kinase with a protein–protein interaction, involved in transcription, translation, or apoptotic processes, and in membrane trafficking and cytoskeletal function [10]. The *LRRK2* and *G2019S* mutations have been reported to be associated with LAS and mitochondrial impairments probably mediated by a gain-of-function effect.

In the same way, deficiency and mutations in VPS35 (the encoded protein is involved in the retromer complex) not only act in the recycling of membrane proteins via retrograde transport from endosomes back to the trans-Golgi (endosome-to-Golgi retrieval) but also have been associated with decreased cellular levels of the lysosome-associated membrane glycoprotein 2A (LAMP-2A), a protein membrane involved in lysosome translocation, affecting once again the LAS [11].

Among the recessive genes linked to PD, proteins encoded by *PARK2* and *PINK1* cooperate in the clearance of damaged mitochondria through mitophagy. Impaired degradation of MIRO (a protein in the outer mitochondrial membrane that connects the organelle to microtubule motors) seems to have a role in defective clearance of damaged mitochondria [6,7,8].

Parkin is an E3 ubiquitin ligase protein, catalyzing the transfer of ubiquitin to its specific target protein; PINK1 is a mitochondrial kinase that localizes to damaged mitochondria and recruits Parkin in the outer mitochondrial membrane to initiate polyubiquitination of mitofusins for fusion and fission of damaged mitochondria or clearance by UBQ-PS or autophagy involving, again, the LAS [12].

Inborn errors of metabolism (IEM) are characterized by mutations in genes coding enzymes involved in different metabolic pathways. Lysosomal diseases enclose an extensive number of genetic disorders characterized by malfunction of the lysosomal enzymes in the LAS [13,14].

Gaucher disease (GD) is the most frequent lysosomal storage disease inherited in an autosomal recessive pattern [4,15,16]. More than 300 different mutations of the glucocerebrosidase 1 (GBA1) gene have been described, with over 12 different genotypes. This gene is located on chromosome 1q2 and encodes glucocerebrosidase (GCase). The GCase enzyme catalyzes the hydrolysis of glycolipid glucocerebroside to ceramide and glucose [17]. It is synthesized in the endoplasmic reticulum (ER) and transported to lysosomes via lysosomal membrane protein 2 (LIMP2). The binding of GCase to LIMP2 is facilitated by the neutral pH of the ER. These proteins remain together at the Golgi apparatus and endosomes, but their dissociation is facilitated by acidic pH into the lysosome [18,19].

Glucocerebroside accumulation results in a systemic disease with distinctive phenotypes [4]. The clinical classification describes three different subtypes, GD 1, 2, and 3, respectively [13,20].

Type 2 or acute neuronopathic is the more severe phenotype and is beyond the scope of the present review, affecting perinatal and infancy with a severe prognosis and limited survival of no more than 3 years, with severe ocular abnormalities, development delay and brainstem involvement and severe hematological and visceral compromise.

Gaucher disease type 3 is called subacute neuronopathic variant with age at onset in childhood with neurological involvement including oculomotor abnormalities, ataxia, seizures (myoclonic epilepsy) and dementia.

Finally, GD type 1 is classically mentioned as non-neuronopathic, with a wide spectrum of age at onset, anemia, thrombocytopenia, and enlargement of the spleen, skeletal abnormalities, interstitial disease and pulmonary hypertension. However, it has been estimated that a neurological symptom occurs in 50% of patients with GD1, and it is possible to identify a neurological abnormality during examination in 30% of patients without neurological complaints [21,22,23].

Epidemiological studies in GD type 1 showed an association between GCase deficiency and Parkinsonism. In fact, the homozygous and heterozygous mutations, constitute a strongest risk factor for the development of PD and Lewy Body Dementia (LBP) [4].

This review primarily focuses on the potential link between GCase deficiency and PD and identifies new potential common pharmacological approaches to GD, a rare treatable disorder, and PD [24].

## 2. Methods

### 2.1. Search Strategy and Selection Criteria

A literature search was conducted to identify relevant articles published in English, based on Medline (via Pubmed) from January 2008 to December 2018. One local article in Spanish was included [25].

### 2.2. Lysosomal Diseases and Gaucher Epidemiology

Several movement disorders have been described in the spectrum of lysosomal diseases, among them levodopa responsive parkinsonism and parkinsonism plus (ataxia, dystonia or spasticity) [4,13,21,26]. In this sense, *GBA* mutation, neuronal ceroid lipofuscinosis, Kufor–Rakeb disease, Niemann Pick type C are among the lysosomal disorders associated to hypokinetic movement disorders and more specifically to Parkinsonisms [21].

In a recent study of a cohort of 76 individuals with different lysosomal diseases, GD was identified in 3.99% of them [26].

Among the conferred risk of genes and genetic loci associated with the development of idiopathic PD, it was observed that *GBA* mutations are the most common genetic risk factor for developing PD [27,28]. Furthermore, GD patients and GBA mutation carriers are at higher risk of developing parkinsonism. Large epidemiological studies found that *GBA* mutations were significantly prevalent in PD patients (Odd Ratio: 5.43); between 5% and 20% of all PD patients have a *GBA* mutation [4,29,30,31].

Further, *GBA1* mutations are the most common genetic risk factor for developing PD [27]. The worldwide range of prevalence of GD type 1 has been estimated at between 1:40,000 and 1:60,000, with the highest prevalence in Ashkenazi Jews (1:850) [32] (19.20%), intermediated prevalence in the North American population (12.93%–15.90%), and lowest in the Asian population (2.70%–3.70%) [27,33,34,35,36,37,38,39,40,41,42,43,44,45,46]. This association is stronger for dementia with Lewy bodies (LBD).

Epidemiological data suggest that non-neuropathic GD type 1 needs to be redefined, taking into account the occurrence of neurological signs and symptoms such as movement disorders, cognitive decline, slow saccades and progressive supranuclear palsy [21,47].

### 2.3. Phenotype/Genotype: Clinical Features

From a clinical point of view, some differences arise between idiopathic PD and *GBA* mutated carriers (*GBA*mtt carriers) with PD. For instance, PD-GBA carriers tend to have a younger age at onset. A good response to l-dopa is a common finding; however, there is contrasting evidence for the occurrence of levodopa-induced dyskinesias, and in those cases the risk is related to the age at onset.

Non-motor symptoms, autonomic dysfunction (including enteric, sexual and urinary dysfunctions as well as orthostatic hypotension), fatigue, anxiety, pain, REM behavior disorders (RBD) and cognitive impairment are more frequent in *GBA*mtt carriers than in individuals with idiopathic Parkinsonism.

A multi-domain impairment has been reported involving memory, visuospatial, abstraction, orientation, working memory, executive, visuospatial abilities and visual short-term memory.

Mood, behavioral and psychiatric symptoms appear as a common manifestation in *GBA*-PD, with earlier development of psychosis and hallucinations, as well as higher prevalence of depression, apathy and anxiety [4,48].

### 2.4. Biomarkers

Clinical and neuroimaging biomarkers in *GBA*mtt carriers with PD with respect to the idiopathic PD (iPD) patients are presented in Table 1.

#### 2.4.1. Wet Biomarkers in GBA Mutation Carriers PD

Dried blood spot and cerebrospinal fluid (CSF) studies demonstrated a decreased glucocerebrosidase (GCase) activity in PD patients with and without GBA mutation carriers versus healthy controls. The decreased activity correlates with a worse cognitive performance [54,55,56].

#### 2.4.2. Prodromal Signs in PD-GBA Patients

As in iPD, prodromal signs have been described in GD type 1 patients and *GBA* mtt carriers. Hyposmia, cognitive dysfunction (involvement of attention, working memory and speed of memory), subtle motor signs, depression, smell and autonomic dysfunction were more common in GBA patients and carriers. Thus, these are suggested as potential neurodegeneration markers in GD patients and carriers [35,47].

Gatto et al. conducted a study in the city of Buenos Aires, Argentina where prodromal clinical markers of PD were explored in GD patients [57]. A total of 26 patients with GD1 were included, and all of them were under enzymatic replacement therapy. Questionnaires used to identify non-motor PD symptoms revealed that 26.9% had parasomnias, 7.69% RBD and constipation, 3.84% hyposmia and 11.53% depression. Some 44.4% had some degree of cognitive impairment. Although none of the patients studied fulfilled Queen Square Brain Bank criteria for PD, the presence of non-motor symptoms, as in other series, could be used as potential prodromic biomarkers for Parkinsonism [57]. 

#### 2.4.3. Cognition in GBA Homozygous and Heterozygous GBA Mutations Carriers

Although data on cognition in asymptomatic *GBA* mutation carriers are scarce, several authors found substantially increased risk of conversion to dementia in GBA mutations carriers [58].

GD type1 and GBA mtt carriers were associated with an earlier age at onset of PD and a higher MDS-UPDRS III, associated with attention, working memory and speed memory impairment. In these cases, the cognitive decline represents one of the most debilitating manifestation impairing the quality of life [59]).

Genetic factors could contribute to modulate the risk of PD and dementia in *GBA* carriers. For instance, null/severe L444P mutations have the highest risk, while an intermediate risk has been reported for mild mutations such as N370S, and the lowest risk was associated with a E326K polymorphism.

In a study conducted by Mata et al. in 2016 [60], the authors found that pathogenic mutations and the E326K polymorphism within the *GBA* gene were associated with a higher prevalence of dementia involving working memory/executive function and visuospatial abilities. These results suggest that even homozygous carriers for E326K polymorphism do not develop GD; this single nucleotide polymorphism might influence the risk of PD cognitive dysfunction.

Controversial results were reported when the *LRRK2* and *GBA* gene mutation carrier cohorts were compared. Some authors failed to identify any cognitive difference in asymptomatic *GBA* and *LRRK2* mutation carriers [61], whereas others identified a lower mean MoCA score and a worsening verbal memory in non-manifesting *LRRK2* carriers with respect to the *GBA* mutations carriers [62]. It remains under discussion whether a more diffuse and extensive neocortical Lewy body pathology increases the risk of cognitive dysfunction in homozygous and heterozygous *GBA* mtt carriers. However, only a marginal difference was found in a PD clinic pathological study performed in *GBA* mutation carriers and non-carriers.

Finally, as in iPD, depression in *GBA* carriers appears as a prodromal factor influencing the performance in cognitive testing.

#### 2.4.4. The Role of Autophagy in Lysosomal Diseases and Neurodegeneration (α-synuclein)

An extensive number of experimental studies showed that *GBA* can stabilize α-synuclein oligomers which in turn inhibit *GBA* function, causing glycocylceramide (GlcCer) accumulation and further attenuate α-synuclein aggregation [63].

Under normal conditions, the autophagy system allows the cell to degrade different compounds. The different types of autophagy are: Microautophagy, macroautophagy and chaperone-mediated autophagy (CMA). These are carried out differently, but the final common pathway is the lysosome, a key player in proteins, lipids and organelles degradation [64,65].

We make a special note of alpha synuclein (α-syn), α-syn is a presynaptic protein, involved in neurotransmitter release through the SNARE complex. When an impairment in the degradation of α-syn occurs, this protein accumulates as insoluble fibrils, giving rise to toxicity in multiple cellular processes (lysosome, mitochondria, proteasome and cellular membrane recycle) [66,67].

It is thought that this accumulation is derived from Lewy body pathology. However, some authors propose a protective cycle through protein accumulation mediated by α-syn. Interestingly, α-syn accumulation leads to reduced GCase and GCase accumulation makes the cell prone to α-syn deposition. Thus, this pathological cycle between GCase and α-syn worsens the condition [4,29].

Cellular protein accumulation promotes what is known as endoplasmic reticulum stress (ERS). When activated, it leads to an apoptotic pathway. Moreover, when ERS is activated there is an inhibition of other ER substrates as well as malfunction of Golgi traffic. This process has also been observed in PD patients with *PARK2* mutations, thus suggesting that ERS has a role in PD pathology [4,29].

#### 2.4.5. *GBA* Gene Mutations

Next-generation sequencing technologies have had a dramatic impact on the field of genomic research and on knowledge of *GBA* mutations. This autosomal recessive disease is caused by different mutations in the *GBA* gene that encode lysosomal enzyme glucocerebrosidase (Gcase), (in chromosome 1q21 [68]). This gene contains 11 exons and 10 introns, covering 7.6 kilobases (kb) of sequence. Over 300 mutations, including point mutations, insertions, deletions and frameshift mutations, in the *GBA* gene have been identified; seven of them account for approximately 96% of the mutant alleles in Ashkenazi Jews (AJ). The most common mutations are: K198T, E326K, T369M, N370S, V394L, D409H, L444P, and R496H. Both N370S and R496H are considered mild mutations, whereas E326K, N370S, and L444P are associated with severe neuronopathic forms of GD. The most deleterious is considered to be L44P, causing high protein destabilization, related to its position at the beginning of the beta sheet [4,16,68,69].

The most common mutation in the *GBA* gene worldwide is N370S/N370S, followed by N370S/L444P [70]. Severe *GBA* mutations (L444P) cause neuronopathic GD onset during infancy and childhood, rapid progression, severe neurological symptoms and shorter life expectancy. Mild GD is caused by N370S mutations; interestingly over 50% of GD-PD are homozygous for these mutations, and 90% of these patients carry at least one N370S mutation. This is important to take into consideration for carriers of *GBA* mutations where severe mutations are related to a higher risk of developing PD7. Moreover, mutations in *GBA* coding for pathogenic neuropathic GD and heterozygous severe forms accelerate cognitive decline in these patients [71].

The increased PD risk in *GBA* mutation carriers is racially dependent. The analysis in AJ population identified 84 insGG and R496H variants as the exclusive risk for increased PD in this population, whereas, in non-AJ, L444P, R120W, IVS2+1G > A, H255Q, D409H, RecNciI, E326K, and T369M represent the highest risk variants with an ethnic distribution. The N370S appears as a risk variant of PD in AJ and non-AJ populations, while L444P increased the risk of PD in all groups in non-AJ ethnicity [72]. Other variants, including N370S, H255Q, D409H and E326K, exclusively increased PD risk in non-AJ European/West Asians, whereas R120W increased PD risk in East Asians.

The polymorphic variant E326K represents an interesting variant to analyze, taking into account that controversial results have been reported regarding the risk of PD. A recent meta-analysis reveals that E326K of *GBA* is associated with a risk of PD in total populations, Asians and Caucasians [73,74].

In the Argentinean GD population, the prevalent variants were: Genotype N307S/other allele (82.5%), N307S/L444P and N307S/N307S [25].

A correlation genotype/phenotype is presented in Table 2.

### 2.5. Gaucher and PD: the Ethiopatogenic Link

Parkinson’s disease is the second most common neurodegenerative disorder worldwide. Multiple pathways for cortical and subcortical structures are involved in the pathology. The hallmark for PD is the intracellular aggregation of α-synuclein. As discussed earlier, PD emerges as a consequence of the failure of multiple cellular pathways to avoid damage by toxic protein accumulation. The failure of protein homeostasis leads to dysfunction of the two major catabolic pathways, UPS, and the autophagy-lysosomal pathway (ALP), as well as mitochondrial, ER and vesicular transport [75,76].

Alpha-synuclein constitutes the major component of LB. It has been proposed that LB deposition follows a sequential pattern of accumulation, as proposed by Braak et al. in 2003 [77]. Initially affecting the dorsal nucleus of the glossopharyngeal and vagal nerves, brainstem, mesocortex and lastly neocortex [4].

Early studies demonstrated a colocalization of mutant GCase in LB and Lewy neuritis (LN) in subjects carrying *GBA*.

Moreover, several recent studies have shown that the levels of GCase catalytic activity is reduced in *GBA* homozygous and heterozygous carriers as well as mRNA GCase levels. The decreased GCase activity was not restricted to *GBA* carriers but was also identified in iPD and DLB, with a marked distribution in different brain areas and more pronounced in Substantia Nigra (SN) [1].

The decrease of GCase activity correlates with *GBA* post translational regulators, protein interactors, lysosomal integral membrane protein 2 (LIMP-2) and saposin C (SapC) [1].

Several GCase mutations, including N370S and L444P, unfold in the ER, activating the unfolded protein response (UPR).

Pathways of GCase from the ER to the lysosome in wild and mutant GCase are presented in Figure 1.

Mutant GCase (mtt GCase) leads to ER stress (ERAD), inducement of UPR, proteosomal breakdown by UBQ-PS, cytoplasmic chaperone-mediated autophagy (CMA), delivery of unfolded proteins into the lysosome by chaperones, and the involvement of LAMP-2A, a protein membrane, in translocation. Glycocylceramide accumulates in the lysosome. Macroautophagy is involved in the degradation of damaged organelles and aggregated proteins and modified lipids [78].

The GCase needs to interact at the ER with LIMP-2 to be glycosylated and transported to lysosome to exert hydrolytic activity on GlcCer [79].

### 2.6. α-Synuclein and GCase Link

The PD pathophysiologic mechanisms are very complex, involving several pathways related to a failure of α-synuclein degradation, oxidative stress, neuroinflammation, and mitochondrial and synaptic dysfunction. Ubiquitin proteasome dysfunction, macroautophagy and CMA impairment promote α-synuclein aggregation and a prion-like transmission.

Both PD and GD share pathological processes that result in lysosomal dysfunction, dysfunctional lipid metabolism, prion-like transmission and bidirectional feedback loop. As a result of the incomplete clearance of these substrates, in GBA-PD, a decrease in GCase activity results in increased levels of glucosylceramide, affecting autophagy and promoting α-synuclein accumulation by stabilization of α-synuclein oligomeric forms [80]. High levels of intracellular α-synuclein prones subsequent ERAD and contributes to GCase glycosylation as well as trafficking dysfunction from ER to Golgi and finally to lysosomes. This pathological loop enhances, so accumulation of glucosylceramide causes α-synuclein, and high α-synuclein levels inhibit GCase. The final consequence is a loss of lysosomal activity and neuronal death [4,81].

Recently, Thomas et al. [82] identified a membrane lipid composition alteration in Drosophila mutants with deletions in the *GBA* ortholog Gba1b. This membrane alteration increases the formation and release of extracellular vesicles that might lead to aggregates seeding and spread cell-to-cell neurodegeneration as a major mechanism for the association of *GBA* and PD neurodegeneration.

### 2.7. Parkin-Pink1 Mitochondria and GCase

Dysfunctional mitochondria and failure in mitophagy (macroautophagy) have been identified in brain tissues from GBA-PD patients and *GBA*L444P.

Furthermore, *SNCA*, *PINK1* and *PRKN*, *PARK7* and *LRRK2* have a role in the equilibrium between mitochondrial fusion and fission [83,84,85,86,87,88,89].

Li et al. reported, in an experimental model of GBA-PD, two mechanisms affecting mitochondria: (a) The impairment of autophagy secondary to lysosomal accumulation of glucosylceramide with decreased GBA activity, and (b) mitochondrial priming, with decreased mitochondrial fission [88,90].

The mitochondrial priming represents the *PINK1-PARK2* pathway required for the balance between fusion and fission [88].

The interaction between Parkin2 and GBA is restricted not only to mitochondrial involvement but also by competitive ubiquitination of mutant GCase, promoting protein accumulation, leading ERAD, increasing cytosolic Ca^2+^ and apoptosis [89].

### 2.8. Therapeutic Implications

Amongst the treatment strategies in patients with GD, either substrate reduction therapy (SRT) or enzyme replacement therapy (ERT) is traditionally employed. For the former, the target is to inhibit glucosylceramide synthase.

When considering ERT, imiglucerase (effective for GD1 patients), velaglucerase alfa and taliglucerase alfa are the available options. None of these is able to cross the blood–brain barrier (BBB), being ineffective in neurological symptoms [4].

Substrate reduction therapy is considered second in line for the treatment of GD, because of its adverse events. Some drugs are able to cross the BBB, such as miglustat, a small iminosugar with reversible inhibitor activity. It was thought that it could be useful for GD3 patients; however, a randomized study did not prove significant difference in the patients in terms of neurological symptoms [4,91].

Due to the relationship between GCase and α-synuclein deposition, new promising therapies are under investigation for patients with PD and GBA mutations. In this matter, the MOVES-PD trial has been announced, in which GZ/SAR402671 will be tested in PD patients with a single GBA mutation in order to reduce the production of glycosphingolipids [92]. On this point, Sardi et al. [93] showed, in experimental models, that α-syn accumulation could be reduced using a glycosylceramide synthase inhibitor (GCC) called Venglustat. It was demonstrated that GCC could reduce the levels of glycosylceramide in the central nervous system (CNS), reduce the accumulation of α -syn in the hippocampus and ameliorate cognitive deficits, making this a promising disease-modifying therapy.

### 2.9. Future Therapies for GD

Due to the multiple cellular pathways involved in GD, other therapies that target different sites of these pathways are under investigation. Chaperones are small molecules that facilitate the correct folding and translocation of GCase, hence making them a suitable option for treating lysosomal disorders. An example of this is ambroxol (ABX).

Chaperones bind to misfolded GCase and cross BBB. Ambroxol acts as a pharmacological chaperone, enhancing lysosomal function and autophagy. It has been shown that ABX significantly increases glycosylceramidase and reduces α-syn, especially in the striatum. Antioxidative functions of ABX have also been postulated as an important property [29,94]. A novel non-inhibitory GCase chaperone, NCGC607, restored the levels of GCase activity and reduced α-syn levels in dopaminergic neurons [95].

Other therapies included histone deacetylase inhibitor, promoting the activity of the mutant GCase [96]. Also, lentiviral vectors with cellular promoters may play a role in future clinical gene therapy protocols for GD1 [96]. Autophagy enhancement through the mTor-pathway, using rapamycins, has been shown to reduce α-synuclein aggregation [4].

Recently, Zunke et al. [97] demonstrated that accumulation of glycosphingolipids in GD promote conformational changes in α-synuclein-leading aggregation and toxicity. In this scenario the reduction of glycosphingolipids appears as a potential new therapeutic pathway, taking into account the fact that this reduction was able to reduce pathology and reverse α-synuclein to the normal conformation in carrier and non-carrier PD patients.

More recently, Kim et al. [98] suggested a new therapeutic approach by inhibition of acid ceramidase. This inhibition helps increase the ceramide levels in lysosome in GCase mutant cells and reduce α-synuclein accumulation.

## 3. Conclusions

As previously mentioned, *GBA* mutations are the most common genetic risk factors associated with PD, especially common in AJ populations. Multiple cellular pathways are linked to GD. This includes lysosomal dysfunction, ERS, autophagy and α-syn deposition, each of them enhancing a vicious cycle of more protein misfolding and deposition. As proposed by Espay et al. [99]. Parkinson’s disease could be considered as a group of disorders that share nigral dopamine-neuron degeneration; hence, PD is divided into different subgroups of PD with their own distinctive biology. This could be useful in the development of disease-modifying therapies for each subgroup of targeted patients. Parkinson’s disease in GD patients could be a subgroup of patients for whom disease-modifying therapies that reduce α-syn and slow, reduce or even stop disease progression could be effective. More clinical trials are required in order to analyze these patients.

## Figures and Tables

**Figure 1 brainsci-09-00030-f001:**
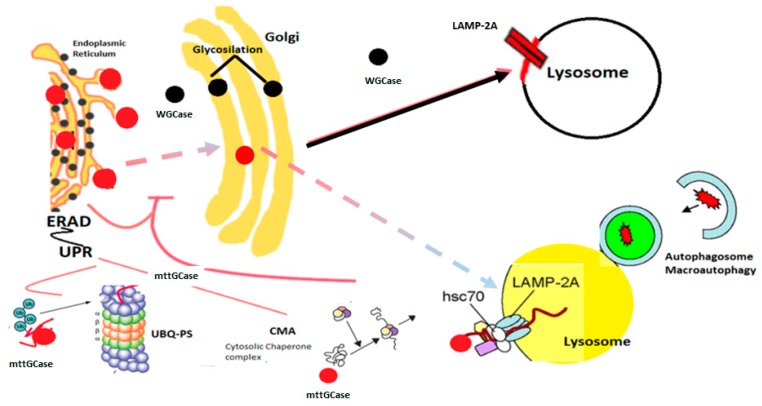
Glucocerebrosidase pathway: Black circles represent wild-type glucocerebrosidase (wtCGase) that is produced in the endoplasmic reticulum (ER), glycosylated in the Golgi, and is translocated to the lysosome in a LIMP-2 dependent process, where it degrades glucosylceramide substrates. Red circles represent mutant enzyme (mttGCase), not folded correctly and inducing the ER stress response. This ER stress response comprises: The ER-associated protein degradation (ERAD) that re-translocated mttGCase from the ER to the cytoplasm and unfolded protein response (UPR) in an attempt to re-establish homeostasis via ubiquitin proteasome system (UBQ-PS), cytosolic chaperone complex (CMA) represents another pathway to refold mttGCase by hsc70 linked to the LAMP-2A to deliver the protein to the lysosome. The dotted line represents the small fraction of mttGCase that could take the normal pathway.

**Table 1 brainsci-09-00030-t001:** Clinical and neuroimaging potential biomarkers of Gaucher disease (GD) type 1.

Biomarker	Observation	References
Clinical biomarker	Early multidomain cognitive impairment. More severe Levodopa induced dyskinesias.	
Transcranial sonography	Nigral hyperechogenicity.	[49]
PET 1 8F dopa	Decreased striatal dopamine synthesis, similar to iPD. Bilateral asymmetric reduction in striatal uptake.	[50,51]
fMRI	Significant hypometabolism in glucose metabolism in supplementary motor area and parieto-occipital cortices. Hypermetabolism of the lentiform nuclei and thalamus. Decrease in the parieto-occipital and to a certain degree anteromedial frontal cortex.	[52]
Diffusion tensor MRI	Decreased frontal cortico-cortical and parahippocampal tracts in GBA-PD. Decreased fractional anisotropy of the corpus callosum, olfactory tract, anterior limb of the internal capsule, cingulum, external capsule bilaterally, and left superior longitudinal fasciculus.	[52]
Postsynaptic DA 11 C-Raclopride	Postsynaptic dopamine terminal persistence of higher putaminal uptake in advanced disease.	[53]

GD: Gaucher Disease. PET: Positron Emission Tomography. GD-PD: Gaucher Disease–Parkinson Disease. iPD: idiopathic Parkinson Disease. fMRI: functional magnetic resonance imaging. MRI: magnetic resonance imaging. DA: dopamine.

**Table 2 brainsci-09-00030-t002:** Phenotype/genotype correlation.

	Null or Severe GBAmtt	Mild GD
	L444P	N307S
Phenotype	Onset infancy and childhood, rapid progression shorter life expectancy, and appearance of more severe neurologic features (GD2, GD3)	50% GD-PD homozygous for N307S 90% GD-PD carry at least one N307S mutation

For *GBA* mutation carriers, “severe” mutations have a higher risk of Parkinson’s disease (PD) than “mild mutations,” as well as early age onset of symptoms, initial bradykinesia and family history of dementia [14,62]. GD: Gaucher disease. *GBA*: glucocerebrosidase. GD-PD: Gaucher-DiseaseParkinson Disease.

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
