# Peer review of "Parkinsonisms and Glucocerebrosidase Deficiency: A Comprehensive Review for Molecular and Cellular Mechanism of Glucocerebrosidase Deficiency"

_brainsci, 2019, doi:10.3390/brainsci9020030_

Round 1
Reviewer 1 Report
In this review article the authors gives an overview of the autophagic-lysosomal dysfunction, in Parkinson’s disease, Gaucher disease and glucocerebrosidase, and is especially focused on the Glucocerebrosidase mutations. I have the following comments:
1. Title: brief and comprehensive seems to be contradictory words.
2. The review is based on data searching January to December, 2018. This period of time seems to be very short in order to summarize the field.
3. The relevant findings presented in the abstract doesn’t fully coincide with the findings presented in the review.
4. Line 53, give example of genes altered in PD. It would be valuable to add more information regarding the proteostasis of alpha-synuclein already in the introduction.
5. Please write all gene names in italic text.
6. Dysfunctional proteostasis would maybe be a better term to use than loss of function.
7. Line 95, were all papers from non-English language peer reviewed journals included?
8. Similar content in the sentences at line 115-116 and line119-121.
9. Please check all abbreviations.
10. Lacking references in the section (line 197-200).
11. Figure 1 Please write a more descriptive text including all concepts (e.g. ERAD, UPR, and UBQ-PS) presented in the picture. Is not mutated CGase transported through the Golgi Network? Is it necessary to use the term wild-type?
12. Figure 2. This figure is very confusing and could be improved. Please also write a more comprehensive figure text.
Author Response
Reviewer 1:
1. Title: brief and comprehensive seems to be contradictory words.
The title was changed to “Parkinsonisms and Glucocerebrosidase deficiency: a comprehensive review”
2. It was based on data searching from January to December, 2018. This period of time seems to be very short in order to summarize the field.
A literature search was conducted between January 2008 and December 2018. The mistake was corrected in the manuscript.
3. The relevant findings presented in the abstract doesn’t coincide exactly with the findings presented in the review.
The manuscript has been restructured to match with the abstract
4. Line 53, give example of genes altered in PD. It would be valuable to add more information regarding the proteostasis of alpha-synuclein already in the introduction.
Examples of genes altered in Pd were added. More information regarding proteostasis of alpha –synuclein was included in the introduction.
5. Please write all gene names in italics.
The manuscript was reviewed and the changes on genes names were corrected.
6. Dysfunctional proteostasis may be a better term than loss of function.
“New genetic findings contribute to support a concomitant loss of proteostasis involving several (…)”
This line was replaced by “New genetic findings contribute to support a concomitant dysfunctional proteostasis involving several (…)”
7. Line 95, all the papers, from other Languages than English and peers journals were included.
The selection only includes relevant articles published in English language peer-reviewed journals. The mistake was corrected in the manuscript.
8. Similar content in line 115-116 and line 119-121.
The text was modified
9. Please check all abbreviations
The manuscript was reviewed and abbreviations corrected.
10. Lacking references in the section (line 197-200)
References to the lines mentioned were added
11. Figure 1. Please write a more descriptive text including all concepts (e.g. ERAD, UPR, and UBQ-PS) presented in the picture. Is not mutated CGase transported through the Golgi Network? Is it necessary to use the term wild-type?
Figure 1 was restructured and rewritten, including responses to the questions above.
12. Figure 2. This figure is very confusing and should be improved. Please also write a more comprehensive figure text.
Figure 2 was removed from the original manuscript to avoid duplication of information in Fig1 legend..
Reviewer 2 Report
In the review ““Parkinsonisms and Glucocerebrosidase deficiency: a brief and comprehensive review for new therapeutic targets” the authors review the recent scientific research, focusing on the link between GCase deficiency and Parkinsonism.
There are several issues in the current form of the review:
1) The title of the paper is misleading – while the aim is to review new therapeutic targets for GCase deficiency mediated Parkinsonism, the review deals with this aspect only briefly in the end of the text. Therefore, the review should be restructured, highlighting the molecular and cellular mechanism of GCase deficiency (mentioned in p.7-9), and their potential therapeutic implications.
2) The language, punctuation and formatting must be corrected throughout the entire text.
3) Material and methods section should be removed from the text, as well as from the abstract.
4) Relevant information from the last year is missing (e.g. Zunke et al. Neuron 2018, Kim et al. HMG 2018, Papadopoulus et al. Hum Genet Metab 2018 ....).
Author Response
Reviewer 2
1) The title of the paper is misleading – the aim is to review new therapeutic targets for GCase deficiency mediated Parkinsonism, and it only refers to this aspect briefly in the end of the text. Therefore, the review should be restructured, highlighting the molecular and cellular mechanism of GCase deficiency (mentioned in p.7-9), and their potential therapeutic implications.
The title of the manuscript was re written as follows: “Parkinsonisms and Glucocerebrosidase deficiency: a comprehensive review. ”Review was restructured to focusing on molecular and cellular mechanism of GCase deficiency
2) The language, punctuation and formatting must be corrected throughout the entire text.
English was edited and reviewed.
3) Material and methods section should be removed from the text; also from the abstract.
Changes were made accordingly.
4) Relevant information from last year is missing (e.g. Zunke et al. Neuron 2018, Kim et al. HMG 2018, Papadopoulus et al. Hum Genet Metab 2018 ..).
References to Zunke et al. 2018 as well as Kim et al. 2018 were included. We could not find references of Papadopoulus in Medline database by Pubmed.
Round 2
Reviewer 2 Report
The review has been substantially improved. However, minor issues still remain:
1) There are several syntax, grammar and punctuation errors within the text. For example:
Line 52: “...new criteria diagnosis” should be a new sentence (otherwise a connection to a previous statement should be made).
Line 58: remove the second “this”.
Line 65: remove the “,” in references 6-8.
Line 311, 321: GCase instead of CCase.
Line 339-340: the sentence is not clear.
Line 341: “loss of” instead of “loss in”
These are only few of the many mistakes. Please correct the manuscript thoroughly.
2) Figure 1: please provide a title to the figure (should appear as the first line within the legend.
Author Response
PD is the second most common neurodegenerative disorder. New criteria diagnosis having been redefined in 2015 by Postuma et al, with multiple genes identified as causative or as an increased risk factor [2].
A key player in this matter is the lysosomal-autophagy- system (LAS), being an important target for many of new therapeutic targets in current clinical trials [1].
are Parkin, PINK1, and DJ1, as well as GBA) [6-8].
Mutant GCase (mtt GCase) leads ERstress (ERAD
α-Synuclein and GCase link
High levels of intracellular α-synuclein prones a subsequent ERAD and contributes to a GCase glycosylation as well as a trafficking dysfunction from ER to Golgi and finally to lysosomes
The final consequence is a loss of lysosomal activity and neuronal death [4, 81]
Fig 1. Glucocerebrosidase Pathway: